# The Role of Tumor Necrosis Factor Alpha Antagonists (Anti TNF-α) in Personalized Treatment of Patients with Isolated Polymyalgia Rheumatica (PMR): Past and Possible Future Scenarios

**DOI:** 10.3390/jpm12030329

**Published:** 2022-02-22

**Authors:** Ciro Manzo, Elvis Hysa, Alberto Castagna, Marco Isetta

**Affiliations:** 1Azienda Sanitaria Locale Napoli 3 Sud, Internal and Geriatric Medicine Department, “Mariano Lauro” Hospital, Rheumatologic Outpatient Clinic, 80065 Sant’Agnello, Italy; 2Laboratory of Experimental Rheumatology and Academic Division of Clinical Rheumatology, Internal Medicine Department, San Martino Polyclinic, University of Genoa, 16132 Genoa, Italy; elvis.hysa@gmail.com; 3Azienda Sanitaria Provinciale Catanzaro, Primary Care Department, 88068 Soverato, Italy; alberto.castagna@tiscali.it; 4Library and Knowledge Services, Central and North West London NHS Foundation Trust, London NW1 3AX, UK; marco.isetta@nhs.net

**Keywords:** polymyalgia rheumatica, anti TNF-a, infliximab, etanercept, tumor necrosis factor alfa, interleukin-6, nnarrative review

## Abstract

Background: Glucocorticoids (GCs) are the cornerstone of polymyalgia rheumatica (PMR) therapy, but their long-term use (as is usually necessary in PMR patients) can induce many adverse events. Alternatives have long been sought. The primary aim of our narrative review is to provide an overview about the use of anti-tumor necrosis factor alpha (TNF-α) drugs in patients with PMR, and discuss advantages and disadvantages. Materials and methods: we performed a non-systematic literature search (PRISMA protocol not followed) on PubMed and Medline (OVID interface). Results and Conclusions: only two anti TNF-α drugs have been prescribed to PMR patients: infliximab in 62 patients and etanercept in 28 patients. These drugs were normally used in addition to GCs when significant comorbidities and/or relapsing PMR were present; less commonly, they were used as first-line therapy. In general, they have been scarcely successful in patients with PMR. Indeed, randomized controlled trials did not confirm the positive results reported in case reports and/or case series. However, an administration schedule and study design different from those proposed in the past could favour new scenarios in the interest of PMR patients.

## 1. Introduction

Polymyalgia rheumatica (PMR) is considered to be one of the most common inflammatory rheumatic diseases occurring in older adults [1,2,3]. Its onset peaks in the age group 71–80 years and its prevalence increases until the age of 90, with a slight decrease thereafter. Its annual incidence rate is estimated between 0.12 and 2.3 cases/1000 (depending on study design and population) in persons aged 50 years and older (age 50-plus is a diagnostic criterion) [4,5,6,7]. 

The typical presentation of PMR involves a sudden-onset and disabling pain in both the shoulders and pelvic girdles associated with morning stiffness lasting > 45 min. Symptom onset is so sudden that the patient usually remembers the exact day and hour when the pains started. Nech aching and constitutional symptoms can be additional manifestations [8,9,10]. Inflammatory markers (such as erythrocyte sedimentation rate [ESR] and C-reactive protein [CRP] concentrations) are usually raised at the time of diagnosis or when PMR relapses. Interleukin-6 (IL-6) is considered to be the main cytokine triggering the systemic acute phase reaction in these patients, but its examination is yet to be routinely carried out [11,12]. Giant cell arteritis (GCA), a granulomatous vasculitis affecting aorta and its branches, can be associated with PMR in so-called “Horton’s disease” [13,14,15]. 

A fast remission following <20 mg/day prednisone has been proposed as a diagnostic criterion [16,17,18], and is commonly used to confirm diagnosis in everyday clinical practice. However, even typical PMR patients may vary in their response to low-dose prednisone. For instance, according to the 2012 European League against Rheumatism-American College of Rheumatology (EULAR-ACR) collaborative study, only 71% of patients had a complete glucocorticoid (GC) response at four weeks with an increase to 78% at 26 weeks. Consequently, this study concluded that response to GCs could not be used as part of PMR classification [19]. Still under discussion—due to the dearth of clinical trials and inconclusive results—are initial GC dose, subsequent tapering regimen as well as total duration of therapy [20,21,22]. As a rule, treatment of these patients should be individualized using the minimum effective GC dose and treatment duration. In particular, according to the 2015 EULAR-ACR recommendations, treatment should start with oral GCs at a daily dose between 12.5 and 25 mg prednisone (or equivalent); and corticosteroid course should last at least one year [23]. However, since approximately 30% to 50% of patients manifest spontaneous exacerbations, the corticosteroid regime may need to be extended. Long-term GC therapy can induce many adverse events: diabetes mellitus, hypertension, cataract and osteoporosis are among the most studied ones [24,25,26]. On the contrary, all their emotional and psychological consequences are barely explored, yet they influence the relationship between PMR patient and rheumatologist [27,28]. 

Therefore, alternatives to the use of GCs have long been sought, and tumor necrosis-alpha inhibitors (anti TNF-α) among these. Indeed, the involvement of TNF-α in PMR has been documented, and some investigators have reported its increased levels in the interstitium of affected muscles in PMR patients [29]. TNF-α is a protein mainly produced by activated macrophages and T-lymphocytes, and is a key mediator of both acute and chronic systematic inflammatory reactions, triggering a series of various inflammatory cytokines and chemokines [30]. In particular, TNF-α stimulates the production and induces the secretion of IL-6, whose over-production is a typical finding in patients with PMR [31,32]. An inappropriate and/or excessive TNF- α signalling is common in some inflammatory and/or autoimmune diseases. The strategies developed by pharmaceutical industries to block TNF signalling either utilize anti-TNF antibodies to neutralize soluble TNF (Infliximab, Adalimumab and Golimumab), or Fc fragments of human immunoglobulin 1 fused with extracellular domains of TNF receptor 1 (TNFR1) or 2 (TNFR2) to block the interaction of TNF with TNF receptors (Certolizumab and Etanercept). 

AIM: the primary aim of our paper is to provide an overview about the use of anti TNF-α drugs in patients with PMR, and discuss advantages and disadvantages. The secondary aim is to discuss the administration schedule and study design different from those proposed in the past. 

## 2. Materials and Methods

We performed a non-systematic (PRISMA protocol not followed) literature search on PubMed and Medline (OVID interface) with the following MeSH terms: [*polymyalgia rheumatica* AND *TNF-alpha inhibitors]*; *TNF-alpha*; *infliximab*, *etanercept*, *certolizumab*, *adalimumab*, *golimumab.* The search was carried out on 12 December 2021. Each paper’s reference list was scanned for additional publications meeting this study’s aim, and when papers reported data partially presented in previous articles we opted to choose the most recent published data.

## 3. Results

Only two anti TNF-α drugs have been prescribed to PMR patients: infliximab (62 cases) and etanercept (28 cases). 

**3A-Infliximab in PMR**: infliximab is a chimeric (murine-human) monoclonal anti TNF-α antibody that is able to bind to both soluble and membrane-bound TNF-α with high affinity, blocking their bindings to specific receptor and therefore their biological activity. 

A 2003 study described a pilot treatment performed on four patients with longstanding PMR that had remained active despite GCs treatment, and who had experienced multiple vertebral fractures [33]. In particular, all these patients had relapsed every time the prednisone dose was reduced to 7.5–12.5 mg/day. They received three intravenous infusions of infliximab 3 mg/kg at weeks 0, 2, and 6 (agreed administration schedule for patients with rheumatoid arthritis [RA]). Prednisone was administered during the first 2 weeks at a dosage of 5 mg/day and then withdrawn if clinical remission occurred after the second infusion. Two patients had a complete response to therapy two weeks after the first infusion; a third patient had sustained clinical remission together with normalization of ESR values two weeks after the first infusion, although IL-6 levels remained elevated during the follow-up period. These three patients were symptom-free with normal ESR and CRP at the end of one-year follow-up. The fourth patient required continuous clinical activity, but the steroid dose was halved. Of particular interest was that patients were unlikely to respond to treatment if ESR did not normalize two weeks after the first infliximab infusion. 

A 2005 article described the administration of infliximab as first line therapy in seven PMR patients with severe comorbidity, such as diabetes mellitus or osteoporotic vertebral fractures [34]. In these patients, methotrexate was then introduced (7.5–10 mg per week, oral or intramuscular administration) for maintaining remission. None of the seven patients ever needed steroid therapy during the follow-up period (mean of eight months). The authors concluded that infliximab could be useful as a first line therapy in the treatment of PMR patients with severe co-morbidities. 

Finally, in 2007, a randomized trial enrolled 51 newly diagnosed and prednisone-naive PMR patients from seven rheumatologic Italian centers [35]. The researchers randomly assigned patients at random to receive an intravenous dose of either infliximab or a placebo at the start of the study and then after 2, 6, 14, and 22 weeks. All patients followed a schedule to slowly decrease the steroid dose from 15 mg to 0 mg over 16 weeks. The ratio of patients who were free of relapse at 52 weeks did not differ between groups, and the authors concluded that infliximab was not useful as a supplementary drug. 

In all these studies, patients with associated GCA were excluded. In Table 1 and Table 2, we list the main findings of these three studies. 

**3B-Etanercept in PMR**: etanercept is a recombinant human TNF dimeric receptor fusion protein consisting of the extracellular portion of two receptors fused to the Fc portion of IgG1. 

A 2007 open pilot study reported six patients with long-standing PMR, in which control of disease was possible only using daily prednisone dosages > 7.5 mg. They received subcutaneous etanercept 25 mg twice weekly for 24 weeks, and were followed up for three months. The prednisone dosage was reduced after the first four weeks to 5 mg/day if a clinical remission was attained, and then reduced to 2.5 mg/day at week 12 if clinical remission persisted. In patients with an incomplete response, its dosage was maintained at 5 mg/day. At the 24-week check-up, prednisone was either withdrawn in the presence of complete remission, or maintained at 2.5 mg/day if remission was incomplete. All six patients reduced their daily prednisone doses without relapses. Five patients presented low disease activity at the end of the treatment period, and one of them was able to stop prednisone at the end of the etanercept treatment [36]. 

In a 2010 randomized controlled trial, etanercept treatment was assessed in 20 newly diagnosed and prednisone-naïve PMR patients compared to 20 control subjects [37]. Etanercept treatment was assessed against placebo in a group of 20 patients with PMR, and in an equal sized group of matched control subjects (without PMR). The trial lasted only 14 days, because its primary outcome was to evaluate fast effect (as for GCs) of etanercept on change in PMR-activity score (PMR-AS). Etanercept monotherapy was found to reduce PMR-AS by 24% in newly diagnosed and GC-naïve patients; however, its scores remained significantly higher than in controls. 

Finally, in 2012, two patients with PMR and comorbidities were treated with etanercept 50 mg/week/subcutaneously. One of them, with ESR and CRP not raised at the time of diagnosis, received an initial dose of 10 mg/day of prednisone. When etanercept was added, prednisone was stopped within two months [38]. 

In Table 3 and Table 4, we list the main features of these studies. GCA was an exclusion criterion.

**3C-Safety and adverse effects****:** generally, infliximab and etanercept treatments were well tolerated in PMR patients, especially in the etanercept group where no serious adverse events were reported. The most common adverse event was a rash as a local injection-site reaction; bacterial cystitis, fatigue and flu-like syndrome were less common adverse events [33,34,35,36,37,38]. 

## 4. Discussion

According to our literature search, infliximab and etanercept are the only anti TNF-α drugs used in patients with PMR. To the best of our knowledge, the use of other different anti TNF-α drugs (such as adalimumab, golimumab and certolizumab) have never been reported in the published literature. Infliximab and etanercept were normally used as drugs in addition to prednisone when significant comorbidities and/or relapsing disease were present; less commonly, they were used as first-line therapy. In the first case, they were both able to reduce doses or stop treatment with prednisone. In the second case, etanercept showed no short-term (14 days) effectiveness while infliximab showed medium-term (a few weeks) effectiveness [33,34,35,36,37,38]. 

Results on circulating TNF-α concentration in PMR patients are remarkably contradictory. In most of the studies, they were similar to those of healthy controls; but some investigators reported higher TNF-α serum concentrations in PMR patients when compared to healthy controls [39,40,41,42]. Detectable plasma TNF-α might not represent the concentration of cytokine locally produced at the site of inflammation. As already highlighted, a study documented increased levels of some proinflammatory cytokines (TNF-α among these) in the interstitium of affected muscles in patients with PMR [27]. 

At present, the exact role of TNF-α in the PMR symptoms is under discussion. The possibility that high levels of TNF-α are present in some PMR patients should be explored case by case. Whether these high levels can characterize a specific subset of disease is an interesting question to which, however, our literature search has not been able to give an answer. 

In all patients, the administration schedule was that for patients with RA. RA, especially in its seronegative elderly-onset subset (EORA is the acronym), is the most confounding disease in everyday clinical practice, to the extent that some researchers ponder whether PMR and EORA are one and the same [43,44]. However, differences in their clinical course, specific ultrasound, and pharmacological approach authorize to not consider PMR and EORA as one and the same. For instance, in patients with EORA, GCs should be considered only when initiating or changing DMARD, followed by their rapid tapering [45,46,47]. 

In particular, with reference to cytokine serum levels, Cutolo et al. reported significantly higher serum levels of IL-6 in PMR than in EORA patients, suggesting a stronger inflammatory state in the first [46]. Circadian rhythms of IL-6 justified a modified-release prednisone (administered at 10 p.m.) in PMR patients [48], and TNF-α has a similar circadian rhythm [49,50,51]. Finally, the same group of investigators discussed PMR as a hypothalamic-pituitary-adrenal (HPA) axis-driven disease that cause serum cortisol levels inappropriately low in relation to the inflammatory state [52]. Whilst acknowledging that this connection is also reported in EORA patients, Sulli et al. demonstrated a higher HPA axis responsiveness in PMR patients [53]. In addition, other investigators reported that increased concentrations of TNF-α may enhance activity and responsiveness of HPA axis in PMR patients [54]. Therefore, anti TNF-α can help improve HPA axis dysregulation. 

Another point of discussion is the patients’ stratification. For instance, the ability to elevate the upper limbs (EUL) = 3 according to the PMR-AS at the time of diagnosis, proved to identify PMR patients who were more likely to require additional therapy [55]. Is it possible that a non-homogeneous stratification of PMR patients involved in the published studies might have favoured incorrect conclusions?

In line with all of these observations, future research designs should take into account the use of slightly higher doses in PMR than in RA patients, discuss infliximab/etanercept (and any other anti TNF-α) administration at 10 p.m., and propose a more homogeneous stratification of PMR patients.

A final point needs to be raised. We chose to review and discuss the role of anti TNF-α, and not of all biological disease modyfing anti rheumatic diseases (b-DMARDS). With the exception of tocilizumab (TCZ), data on other b-DMARDS in PMR patients are preliminary. In particular, recent research is studying the effectiveness of rituximab and abatacept in PMR patients. Rituximab is a monoclonal antibody directed against CD20 B-lymphocytes that is able to cause their lysis. Abatacept binds on the CD80 and CD86 molecule, blocking the so-called second signal between antigen-presenting cell and T cells; the final result is that T-cells cannot be activated. First, preliminary results seem interesting, because some patients with PMR respond to these drugs [56,57]. TCZ is the first humanized anti-IL 6 receptor monoclonal antibody, and has a more consolidated experience in PMR patients. TCZ is effective as a steroid-sparring agent in many (but not in all) PMR patients; its monotherapy is still discussed [58,59,60,61]. All these reports confirm that IL-6 cannot be considered as the only mediator involved in PMR pathogenesis [62].

## 5. Conclusions

According to our literature search, etanercept and infliximab have been scarcely successful in patients with PMR, with the exception of a very small subgroup. Indeed, randomized controlled trials did not confirm the positive results reported in case reports and/or case series.

As suggested, a new administration schedule could be assessed in future research, together with a study design different from those proposed in the past, before coming to the conclusion that these drugs are not effective.

According to our literature search, no studies on anti TNF-α in PMR have been published over the last 12 years. We hope that new research will be encouraged in the interest of PMR patients.

## Figures and Tables

**Table 1 jpm-12-00329-t001:** Infliximab in patients with PMR. Main features of the three discussed studies.

First Author	Year	Patients (No.)	F/M	Age	Study Type	Diagnosis	Comorbidities
Salvarani [33]	2003	4	4/0	63–69 years	pilot study	Healey	vertebral fractures
Migliore [34]	2005	7	7/0	65–84	case-series	Healey	DM, vertebral fractures
Salvarani [35]	2007	51	31/20	71 *	randomized trial	Healey	not relevant

F = females; M = males; DM = diabetes mellitus; * Median.

**Table 2 jpm-12-00329-t002:** Infliximab in patients with PMR: administration schedules, follow-up and results.

Administration Schedule	Prednisone	F/A	Follow-Up	Results
3 mg/Kg IV at week 0, 2, and 6	5 mg/day with tapering	A	49 months *	Complete response in two patients (see text)
3 mg/Kg IV at week 0, 2, and 6	no	F	8.14 months *	Clinical improvement in all patients
3 mg/kg IV at week 0, 2, 6, 14, 22	15 mg/day with fast tapering	A	12 months *	Not significant improvement

IV = intravenous; F = first; A = additional. * Median.

**Table 3 jpm-12-00329-t003:** Etanercept in patients with PMR: main features of the three discussed studies.

First Author	Year	Patients	F/M	Study Type	Diagnosis	Comorbidities
Catanoso [36]	2007	6	5/1	pilot study	Healey	Vertebral fractures, DM, hypertension, cataract
Kreiner [37]	2010	10	6/4	RCT	Chuang	Hypertension, hypercholesterolemia
Aikawa [38]	2012	2	2/0	case report	not clear	Vertebral fractures, cataract, htpertension

F = females; M = males; RCT = randomized controlled trial; DM = diabetes mellitus.

**Table 4 jpm-12-00329-t004:** Etanercept in patients with PMR: admisistration schedules, follow-up and results.

Administration Schedule	Prednisone	F/A	Follow-Up	Results
25 mg twice weekly for 24 weeks	see text	A	9.5 months	Improvement with very low doses of prednisone
25 mg biweekly for 2 weeks	no	F	15 days	Modest improvement without prednisone (see text)
25 mg biweekly	10 mg/day	A		Clinical improvement

F = first; A = additional.

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
