# Peer review of "The Role of Tumor Necrosis Factor Alpha Antagonists (Anti TNF-α) in Personalized Treatment of Patients with Isolated Polymyalgia Rheumatica (PMR): Past and Possible Future Scenarios"

_jpm, 2022, doi:10.3390/jpm12030329_

Round 1
Reviewer 1 Report
This narrative review provides an overview about the use of anti tumor necrosis factor alpha (TNF-α) drugs in patients with PMR, and discuss advantages and disadvantages.
It is a well-structured scientific work and is suitable for publication.
Author Response
Dear Reviewer,
we thank You so much for Your comment !
Ciro Manzo, on behalf of the co-authors
Reviewer 2 Report
Dear authors,
Thank you for your effort to review the previous literature on this important and relevant topic. Please find my suggestions below:
- Please add a table under result section describing the selected studies with more details. You may for instance consider to add data on the patients characteristics, how diagnosis was made, baseline lab tests, comorbid diseases, possible concurrent GCA, prednisolon, studies' measured outcomes, definition of remission, etc, while some of them are already discussed in the text. This can also help to compare the results quickly. In this regard, results of the discussed studies could also be shown by figures or compared statistically, depending on the available data.
- One may also consider to discuss the lack of valid outcome measures for PMR/GCA, as PMR-AS has several weak points.
Best regards.
Author Response
Dear Reviewer,
first of all, we thank for Your time and attention.
Your suggestions are helpful for improving the quality of our manuscript. For instance, we add two tables under Results section describing the studies with more details.
Please, read the revised version of our manuscript.
Instead results of the discussed studies cannot be compared statistically (as you suggest) because their study disegn and the low number of the enrolled patients does not allow it.
The lack of valid outcome measures for PMR/GCA ? Could this be an additional discussion point ? We decide not to follow up this your suggestion, because all the patients enrolled in the discussed studies suffered from isolated PMR and GCA was an exclusion criterion. We are sure you will understand.
Best regards.